# Low-Vacuum Filtration as an Alternative Extracellular Vesicle Concentration Method: A Comparison with Ultracentrifugation and Differential Centrifugation

**DOI:** 10.3390/pharmaceutics12090872

**Published:** 2020-09-13

**Authors:** Anna Drożdż, Agnieszka Kamińska, Magdalena Surman, Agnieszka Gonet-Surówka, Robert Jach, Hubert Huras, Małgorzata Przybyło, Ewa Łucja Stępień

**Affiliations:** 1Department of Medical Physics, Marian Smoluchowski Institute of Physics, Faculty of Physics, Astronomy and Applied Computer Science, Jagiellonian University, 30-348 Krakow, Poland; anna.drozdz@uj.edu.pl (A.D.); agnieszka1.kaminska@uj.edu.pl (A.K.); 2Malopolska Centre of Biotechnology, Jagiellonian University, 30-837 Krakow, Poland; 3Department of Glycoconjugate Biochemistry, Institute of Zoology and Biomedical Research, Jagiellonian University, 30-837 Krakow, Poland; magdalena.surman@doctoral.uj.edu.pl (M.S.); malgorzata.przybylo@uj.edu.pl (M.P.); 4Department of General Chemistry, Faculty of Chemistry, Jagiellonian University, 30-837 Krakow, Poland; agnieszka.gonet-surowka@uj.edu.pl; 5Department of Gynecological Endocrinology, Faculty of Medicine, Jagiellonian University Medical College, 31-007 Krakow, Poland; robert.jach@uj.edu.pl; 6Department of Obstetrics and Perinatology, Faculty of Medicine, Jagiellonian University Medical College, 31-007 Krakow, Poland; hubert.huras@uj.edu.pl

**Keywords:** dialysis membrane, ectosomes, exosomes, FTIR, infrared spectroscopy, purification

## Abstract

Recent years have brought great focus on the development of drug delivery systems based on extracellular vesicles (EVs). Considering the possible applications of EVs as drug carriers, the isolation process is a crucial step. To solve the problems involved in EV isolation, we developed and validated a new EV isolation method—low-vacuum filtration (LVF)—and compared it with two commonly applied procedures—differential centrifugation (DC) and ultracentrifugation (UC). EVs isolated from endothelial cell culture media were characterized by (a) Transmission Electron Microscopy (TEM), (b) Nanoparticle Tracking Analysis (NTA), (c) Western blot and (d) Attenuated Total Reflection Fourier-Transform Infrared Spectroscopy (ATR-FTIR). Additionally, the membrane surface was imaged with Environmental Scanning Electron Microscopy (ESEM). We found that LVF was a reproducible and efficient method for EV isolation from conditioned media. Additionally, we observed a correlation between ATR-FTIR spectra quality and EV and protein concentration. ESEM imaging confirmed that the actual pore diameter was close to the values calculated theoretically. LVF is an easy, fast and inexpensive EV isolation method that allows for the isolation of both ectosomes and exosomes from high-volume sources with good repeatability. We believe that it could be an efficient alternative to commonly applied methods.

## 1. Introduction

Extracellular vesicles (EVs) are defined as bilayer cell membrane fragments released into the extracellular space [1]. The number and composition of EVs can vary, depending on the cells they originate from and the physiological or pathological conditions [2,3]. In experimental conditions, EVs are released into cell culture media, producing a conditioned medium [4,5,6,7]. The classification of EVs is still under debate and reaching a common classification is complicated. To solve this awkward situation, MISEV 2018 guidelines were published, providing an extensive and complete characterization of EVs and their classification according to different parameters [8]. Large variation in size, composition and function has been found among the three types of EVs (Table 1).

EVs are widely studied because of their involvement in cell-to-cell communication [12]; tumor progression [13], and their possible application as biomarkers [14] or drug delivery systems (DDS) [15]. While studying the functional differences between the different types of EVs, it is crucial to use an appropriate isolation method that allows for obtaining a homogenous EV population. Unfortunately, this is a challenging task and the contamination of different EV types is very often observed [16].

An additional difficulty is the isolation of EVs from large-volume sources, or a situation when a large number of EVs is required for downstream analysis. In 2016, a worldwide study on applied EV isolation methods was performed among the members of the International Society of Extracellular Vesicles (ISEV) [17]. The study showed that 81% of respondents isolate EVs from conditioned media. Additionally, 71% of respondents isolate EVs from sources with starting volumes exceeding 5 mL (sometimes even up to 100 mL). According to this worldwide study, the most popular method is ultracentrifugation (81% of respondents), in which the processing of high-volume samples can be difficult because of the relatively small size of the ultracentrifuge tubes usually used. Moreover, ultracentrifugation is a low-yield method and is characterized by high levels of contamination due to the coprecipitation of proteins [8,9,18,19].

Working with high-volume sources such as conditioned cell culture media, we propose a new method of EV isolation that allows for EV concentration in a relatively short time. Low-vacuum filtration (LVF) is a modification of the hydrostatic filtration dialysis (HFD) method described by Musante et al. [20], extended by the application of low vacuum (−0.3 (bar)) in order to obtain faster filtration, which can limit the influence of isolation time on the changes in the sample [21]. The filtration system (Figure 1) consists of a closed cell culture media container from which, through a coupler, cell culture media flows into a dialysis membrane, where filtration is facilitated by the negative pressure in the vacuum chamber generated by the pump. The dialysis membrane is closed with a clamp. During the filtration process, the cell culture media soaks through micropores and EVs are collected within the membrane, which leads to sample concentration. The final volume of a sample may be reduced to 1 (mL)). Thanks to an additional step, membrane washing, the reduction of protein contamination (category 3 proteins [9]) in the sample can be achieved.

The aim of this study was to compare the efficiency of three methods of EV isolation: LVF, differential centrifugation and ultracentrifugation. EVs were isolated from human umbilical endothelial vein cells (HUVEC) conditioned media, and the obtained EV samples were compared in terms of the following parameters: size distribution, morphology, concentration and homogeneity of the EV populations. We also applied Attenuated Total Reflection Fourier-Transform Infrared Spectroscopy (ATR-FTIR) for EV characterization in order to assess biochemical composition to compare the three methods.

## 2. Materials and Methods

### 2.1. Cell Culture

Umbilical cords were collected during C-section performed between the 38^th^ and the 42^nd^ weeks of normal pregnancy and stored in Hank’s balanced salt solution at 4 (°C) until HUVEC isolation. For cell digestion, an umbilical vein was injected with prewarmed (37 (°C)) 0.25% trypsin (cat. No. 85450C, Sigma Aldrich, St. Louis, MI, USA) with EDTA (380 (mg/L)) (cat. No. E6758, Sigma Aldrich, St. Louis, MI, USA) mixed with cell culture medium 199 (cat. No. M7653, Sigma Aldrich, St. Louis, MI, USA) 1:1. Umbilical cords were incubated at 37 (°C) in PBS (phosphate buffered saline) for 30 (min); afterwards, the cells were washed out from the vein, collected in a 50 (mL) Falcon tube and centrifuged at 250 (g) for 15 (min). The cell pellet was suspended in culture medium and cells were seeded into culture flasks.

HUVECs were cultured in a 75 (cm^2^) flask with the 1:1 mixture of 199 medium and human endothelium Serum-Free Medium (SFM; cat. No. 11111044, GIBCO, Gaithersburg, MD, USA), supplemented with 10% Fetal Bovine Serum (FBS; cat. No. S181B-500, Biowest, Nuaillé, France), penicillin/streptomycin (cat. No. P0781 BioReagent, Montigny le Bretonneux, France) at a concentration of 10,000 (units/L) (penicillin) and 1 (mg/L) (streptomycin) and 2 (mM) of l-glutamine (cat. No. 17-605E, BioWhitaker, Lonza, Basel, Switzerland).

Before the sample collection, HUVECs were serum-starved for 24 h to obtain synchronization and avoid contamination by serum EVs [22,23]. Conditioned media were pulled from twelve bottles and divided into three equal portions, one for each method of isolation, to obtain the same starting material composition in a volume of 40 (mL).

### 2.2. Extracellular Vesicle Isolation

Every sample, before the procedure, underwent three preparatory centrifugations (Figure 2). In order to remove intact cells, cell debris and apoptotic bodies, samples were centrifuged successively at 400 (g) (10 (min)), 3100 (g) (25 (min)) and 7000 (g) (20 (min)) at 4 (°C).

#### 2.2.1. Ultracentrifugation (UC)

After the initial centrifugation steps, samples were transferred to 1.5 (mL) top-opened centrifuge tubes and spun for 1.5 (h) at 150,000 (g) at 4 (°C) (Sorvall MX 150+ Micro-Ultracentrifuge, Thermo Fisher Scientific, Waltham, MA, USA). EV pellets were suspended in 50 (µL) of PBS, collected into one tube and spun once again under the same conditions. Samples were prepared in triplicate and pellets were stored at −80 (°C) for downstream analysis.

#### 2.2.2. Differential Centrifugation (DC)

Differential centrifugation was performed according to the previously described protocol [24]. After the three preparatory centrifugations, samples were transferred to 50 (mL) polycarbonate centrifuge tubes and centrifuged for 20 (min) at 18,000 (g) at 4 (°C) (Sorvall LYNX 6000 Superspeed Centrifuge, Thermo Fisher Scientific, Waltham, MA, USA). Part of the supernatant was discarded and the lower part of the medium (1.5 (mL)) was centrifuged in Eppendorf Tubes under the same conditions (5804 R Centrifuge, Eppendorf, Hamburg, Germany). The pellets were resuspended in 1.5 (mL) of PBS and centrifuged again two times under the same conditions (Figure 2). Samples were prepared in triplicate and pellets were stored at −80 (°C) for downstream analysis.

#### 2.2.3. Low-Vacuum Filtration (LVF)

LVF was performed on the dialysis membrane (cat. No. 131486, Spectra/Por Biotech) with MWCO (molecular weight cut-off) 1000 (kDa). The whole system was assembled as presented in Figure 1. After the preliminary centrifugations, 40 (mL) of sample was placed in the liquid container (in Figure 1a), filtered under low vacuum (−0.3 (bar)) and subsequently washed with 15 (mL) of water. After the filtration, the samples, which had been prepared in triplicate, were stored at −80 (°C) for the downstream analysis. For the purpose of TEM imaging, the samples after the filtration were ultracentrifuged under the conditions described in Section 2.2.1.

### 2.3. Environmental Scanning Electron Microscopy (ESEM)

To evaluate the diameter of pores in the dialysis membrane, we applied Environmental Scanning Electron Microscopy (ESEM). A fragment of the dialysis membrane (1 × 1 (cm)) was placed on the SEM sample holder and the subsequent ESEM measurements were performed using the SEM Quanta 3D FEG microscope (FEI Company, Hillsboro, OR, USA) in use by the Department of Solid State Physics (Institute of Physics Jagiellonian University, Kraków, Poland). The ESEM images were collected by a Low-Vacuum Secondary Electron Detector (LVED) using an electron beam of 20 keV energy. During the measurement, the specimen was kept at 130 (Pa) of water vapor at room temperature.

### 2.4. Transmission Electron Microscopy (TEM)

Samples for TEM imaging were prepared as previously described [14]. Pellets of isolated EVs were fixed with 2.5% glutaraldehyde (cat. No. G5882, Sigma-Aldrich, St. Louis, MI, USA) in 0.1 (M) cacodylic buffer (cat. No. C4945, Sigma Aldrich, St. Louis, MI, USA) and then postfixed in 1% osmium tetroxide solution for 1 (h). In the next step, samples were dehydrated in ethanol and embedded in PolyBed 812 (cat. No. 08792-1, Polysciences, Warrington, PA, USA) at 68 °C. Ultrathin sections were placed on the 300 mesh grids, covered with formvar film and contrasted using uranyl acetate and lead citrate. Observations were performed using a JEOL JEM2100HT electron microscope (Jeol Ltd., Tokyo, Japan) with the accelerating voltage of 80 (kV).

Images were analyzed using Photoshop and CTAnalyzer software. Background was removed from the binarized images and EVs as single objects were counted automatically. Four different parameters were considered: diameter, area, solidity and eccentricity. Solidity was calculated according to the following equation:(1)$$solidity=\;\frac{area}{area\;created\;by\;convex\;hull}$$

This is a parameter describing the extent to which a shape is convex or concave. This parameter is equal to 1 for a convex shape with no irregularities, and equal to 0 for a concave shape with many thin insets [25]. Eccentricity was calculated according to the following equation:(2)$$eccentricity=\;\frac{minor\;axis\;length}{major\;axis\;length}$$
which compares the length of the minor axis and major axis, providing information about the changes in the elongation of an object.

### 2.5. Nanoparticle Tracking Analysis (NTA)

NTA measurements were performed by means of the NanoSight LM 10 (Malvern Panalytical, Malvern, UK), coupled with a 405 (nm) laser. For the NTA analysis, 100 (µL) of each sample was diluted to the volume of 500 (µL) with filtered PBS. For each method, three samples were prepared and measured in five independent records for 30 (s). The measurements were analyzed using NTA 3.1. software, and calculated and normalized to the starting sample volumes. The final results were analyzed by means of OriginPro 2018 software.

### 2.6. Attenuated Total Reflection Fourier-Transform Infrared Spectroscopy (ATR-FTIR)

For the ATR-FTIR spectroscopy measurements, 5 (μL) of each sample in PBS was mounted and dried on the diamond crystal of the Nicolet 6700 FT-IR spectrometer (Thermo Fisher Scientific, Waltham, MA, USA) to obtain a thin dry film. Measurements were performed immediately at room temperature and 256 scans were collected at a nominal resolution of 4 (cm^−1^). The analysis of the obtained spectra was performed using the OriginPro 2018 Software.

### 2.7. Electrophoresis and Western Blot

EV protein extracts (15 (μg) per sample) were diluted 1:1 in the Laemmli Sample Buffer (62.5 (mM) Tris-HCl, pH 6.8, 1 (mM) EDTA, 10% glycerol, 2% SDS, 0.025% bromophenol blue with 5% β-mercaptoethanol), separated by electrophoresis using the 4–15% gradient Mini-PROTEAN TGX Stain-Free Protein Gels (cat. No. 4568085, BioRad Laboratories Inc., Hercules, CA, USA) and transferred to PVDF (polyvinylidene fluoride) membranes using the Mini-Protean 3 system (Bio-Rad Laboratories Inc., Hercules, CA, USA).

Western blot analysis was performed using the Lumi-LightPLUS Western Blotting Kit (Mouse/Rabbit) (cat. No. 12015218001, Roche, Basel, Switzerland). The blots were blocked overnight in 1% BSA (bovine serum albumin) in Tris-buffered saline with Tween (0.05 (M) Tris-HCl, 0.15 (M) NaCl, 0.1% Tween 20, pH 7.5), and using the Lumi-LightPLUS Western Blotting Kit (Mouse/Rabbit) were incubated for 1 h with primary antibodies against VCAM (vascular cell adhesion molecule) (dilution 1:500, cat. No. sc-13160, Santa Cruz Biotechnology, Inc., Dallas, TX, USA), Hsp70 (dilution 1:500, cat. No. sc-24, Santa Cruz Biotechnology, Inc., Dallas, TX, USA), Arf- 6 (dilution 1:200, cat. No. sc-7971, Santa Cruz Biotechnology, Inc., Dallas, TX, USA), actin (dilution 1:200, cat. No. sc-47778, Santa Cruz Biotechnology, Inc., Dallas, TX, USA), CD81 (dilution 1:200, cat. No. MABF2061, Sigma Aldrich, St. Louis, MI, USA) and CD63 (dilution 1:500, cat. No. CBL553, Sigma Aldrich, St. Louis, MI, USA). After incubation with the primary antibodies, membranes were washed three times with Tween/TBS buffer and incubated for 1 (h) with an appropriate horseradish peroxidase (HRP)conjugated secondary antibody (Lumi-LightPLUS Western Mouse/Rabbit Blotting Kit) diluted 1:250 in 1% BSA in TBS/Tween buffer. Afterwards, incubation membranes were washed three times in the TBS/Tween buffer and three times in TBS buffer (0.05 M Tris-HCl, 0.15 M NaCl, pH 7.5. Immunopositive bands were visualized using Lumi-Light Reagent (Roche, Basel, Switzerland) and the ChemiDoc™ XRS+ System (Bio-Rad Laboratories Inc., Hercules, CA, USA). The relative levels of protein content were determined using Image Lab software. Individual protein levels were normalized to the total intensity of the bands on a given line, detected in the gel after electrophoresis.

### 2.8. Ethical Statement

The collection of umbilical cords for this study was approved by the Bioethical Committee of Jagiellonian University in Kraków on 26 April 2016 and written informed consent for publication must be obtained from participating patients. A written informed consent for publication had been obtained from participating patients and patient details had been anonymized. Permission number 122.6120.78.2016 was valid until 30 April 2018 and the collection of umbilical cords has been performed within the validity period. 

## 3. Results

### 3.1. Evaluation of Dialysis Membrane Pore Diameter with ESEM

ESEM was used to visualize pores in the dialysis membrane. Figure 3 shows an exemplary image obtained in ESEM. ESEM measurements reveal the irregular structure of the membrane with pores varying in diameter (from 20.59 (nm) to 51.05 (nm)). The average pore size calculated from the 50 randomly selected pores was 28.39 ± 9.63 (nm).

### 3.2. EV Visualization with TEM

TEM was used to confirm the presence of EVs in the analyzed samples. Figure 4 shows representative images of EVs isolated by differential centrifugation, LVF and ultracentrifugation. Additionally, in order to investigate the influence of the applied methods on morphology of EVs, we compared their area, eccentricity and solidity.

The highest numbers of particles detected in the TEM samples were observed for EVs isolated by differential centrifugation (52 particles/image) and LVF (18 particles/image); the number of particles was the lowest for samples isolated by ultracentrifugation (5 particles/image). Additionally, the electron density of EVs was the highest for the LVF method and the lowest for ultracentrifugation. The average diameter of EVs and size distribution varied between samples (Figure 4). The largest EVs, with an average diameter of 227 ± 175 (nm) and a median diameter of 175 (nm), were observed in the samples isolated by differential centrifugation. In samples obtained by LVF, EVs had an average diameter of 114 ± 69 (nm) and a median diameter of 100 (nm). In ultracentrifugation samples, the smallest EVs observed had an average diameter of 78 ± 44 (nm) and a median diameter of 72 (nm). The area of EVs corresponded to the mean diameter and to shape parameters, which were very similar across the analyzed groups (see the table in Figure 4). For each isolation method, we observed the elongation of the EVs: the eccentricity parameter varied between 0.57 ± 0.15 for differential centrifugation and 0.60 ± 0.15 for LVF. We did not observe differences in the solidity of particles. In differential centrifugation and LVF, EVs had the same solidity of 0.92 ± 0.07 and 0.92 ± 0.03, respectively. EVs isolated by the ultracentrifugation method had a solidity of 0.91 ± 0.02.

### 3.3. NTA Measurements of EV Concentration and Size

EV concentrations were measured for the starting samples and for the samples after isolation using NTA. The average concentrations of EVs in the samples after ultracentrifugation was 1.71 × 10^10^ ± 1.23 × 10^8^ (particles/mL). The average size of the EVs detected was 224 ± 112 (nm), and EVs with diameter lower than 100 (nm) were not detected. We observed a 35-fold increase in EV concentration in comparison to the starting sample.

For samples isolated by the LVF method, the average concentration of EVs was 7.96 × 10^9^ ± 5.82 × 10^7^ (particles/mL). The average size was 260 ± 132 (nm), and particles with a diameter lower than 100 (nm) were not detected either. We observed a 22-fold increase in particle concentration in comparison to the starting sample.

The lowest concentrations (4.74 × 10^9^ ± 3.91 × 10^7^ (particles/mL)) were found for samples obtained by differential centrifugation. Average EV size was equal to 255 ± 142 (nm), and unlike the case of the other methods, EVs with a diameter lower than 100 (nm) were detected. We observed a 13-fold increase in EV concentration in comparison to the starting sample. High variations in size and concentration were observed (Figure 5).

### 3.4. Infrared Spectra of EVs

Infrared (IR) spectroscopy provides general information about chemical composition with additional details regarding the quality of the isolated EVs, especially in the context of the protein and lipid content (Table 2). Two amide peaks originating from peptides—the amide I band, at around 1652 (cm^−1^) [26], and the amide II band, at around 1542 (cm^−1^) [26]—were the main component of the EV infrared spectra (Figure 6a,b). Additionally, the band at around the 3286 (cm^−1^) peak belonging to amide A [26] was observed. The highest intensities for the aforementioned peaks, associated with proteins and peptides, were detected for the EVs isolated by LVF and ultracentrifugation. In differential centrifugation samples, those peaks were barely distinguishable, showing low intensity. In the LVF samples, additional peaks were observed at 1309 (cm^−1^) and 1240 (cm^−1^), which were attributed to amide III (C–N stretching mode of proteins).

We observed lipid bands represented as four peaks originating from the stretching vibrations of lipid acyl chain groups (Figure 6a,b). The peaks at 3076 (cm^−1^) and 2959 (cm^−1^) are generated by CH_3_ asymmetric stretching, while the peaks around 2930 (cm^−1^) and 2869 (cm^−1^) are generated by CH_2_ asymmetric and symmetric stretching vibrations. Additional lipid bands originating from CH_2_ [27] and CH_3_ [27] bending vibrations in the lipid acyl chains were distinguished around the 1450 (cm^−1^) and 1397 (cm^−1^) peaks, respectively. As in the case of the protein bands, lipid bands were detected in the LVF and ultracentrifugation samples and were barely distinguishable in the samples of EVs isolated by the differential centrifugation method.

**Table 2 pharmaceutics-12-00872-t002:** FTIR peak assignment.

Wavenumber (cm^−1^)	Definition of the Spectra Assignment
3286	Overlapping –OH stretching vibrations and N–H stretching vibrations from peptide groups of proteins (amide A) [26]
3076 2959	CH_3_ asymmetric stretching vibrations from lipids with low contribution from proteins, carbohydrates and nucleic acids [28]
2930 2869	CH_2_ asymmetric and symmetric stretching vibrations from lipids with low contribution from proteins, carbohydrates and nucleic acids [29]
1652	C=O stretching vibrations from peptide backbone (amide I) [26]
1542	N–H bending vibrations from peptide groups (amide II) [26]
1450	CH_2_ bending (scissoring) vibrations from lipid acyl [27]
1397	CH_3_ bending vibrations from lipids and proteins [27]
1309 1240	C–N stretching mode of proteins, indicating mainly α-helical conformation (amide III) [30]

In order to perform further ATR-FTIR analysis for amide (1450–1750 (cm^−1^)) and lipid (2800–3000 (cm^−1^)) bands, we performed automatic baseline subtraction and Gaussian function fittings to all peaks in the analyzed ranges (Figure 6c). Based on the areas under the curves, we calculated the amide I/lipids ratio and obtained the highest ratio for the LVF and ultracentrifugation samples: 10.22 and 6.31, respectively. IR spectra for EVs isolated by differential centrifugation were characterized with the lowest ratio (4.15) and the total area under the analyzed peaks was much lower than for other samples.

### 3.5. EV Protein Markers

In order to investigate the type of EVs present in the isolated samples, we performed a Western blot analysis. The total protein amount in the EV samples was measured by the BCA method and the highest protein concentration was observed in the LVF samples (3.73 ± 0.63 (mg/mL)). In the ultracentrifugation and differential centrifugation samples, protein amounts were similar: 2.41 ± 1.70 (mg/mL) and 2.40 ± 0.23 (mg/mL), respectively (Figure 7a). To confirm the endothelial origin of the isolated EVs, vascular cell adhesion molecule 1 (VCAM-1) was used as an endothelial marker. High-intensity bands for VCAM-1 were observed in the differential centrifugation and LVF samples: 1.50 ± 0.16 (AU) and 1.38 ± 0.24 (AU), respectively. The intensity of these bands was significantly higher than in the ultracentrifugation samples (0.89 ± 0.11 (AU)) (Figure 7b). Additionally, in the tested samples, actin bands with similar intensity were detected: 1.33 ± 0.33 (AU) for differential centrifugation, 0.88 ± 0.07 (AU) for LVF and 1.19 ± 0.17 (AU) for ultracentrifugation.

We used Arf-6 as an ectosomal marker and detected Arf-6-positive bands with low intensity for all tested samples (0.7 ± 0.02 (AU) for differential centrifugation, 0.5 ± 0.02 (AU) for LVF, 0.7 ± 0.04 (AU) for ultracentrifugation) (Figure 7b). As exosome markers, we used Hsp70, CD63 and CD81. Hsp70 bands with high intensity were detected in the LVF samples (0.48 ± 0.14 (AU)), compared to the ultracentrifugation and differential centrifugation samples: 0.23 ± 0.12 (AU) vs 0.04 ± 0.01 (AU), *p* < 0.05. Bands for the CD81 marker had significantly higher intensity in the LVF (0.23 ± 0.01 (AU)) and ultracentrifugation samples (0.19 ± 0.01 (AU)) compared to the differential centrifugation samples (0.13 ± 0.02, *p* < 0.05). There were no statistically significant differences in the intensity of CD63 band intensity between the methods: differential centrifugation (1.56 ± 0.36 (AU)), LVF (0.96 ± 0.29 (AU)) and ultracentrifugation (0.85 ± 0.22) (Figure 7b).

## 4. Discussion

The interest in extracellular vesicles has grown over the last years [31], mostly because of their involvement in cell-to-cell communication [32], cancer progression [33] and immunosuppression [34], and due to their great potential as DDS [15,35]. Besides the recent intense development in EV isolation micromethods, there is an unmet need to develop more efficient and repeatable EV isolation and concentration methods for application with high-volume sources, suitable not only for proteomic analysis but also for drug delivery purposes [35,36]. In this study, we have developed and validated a system that is dedicated to EV concentration from high-volume sources of conditioned media: low-vacuum filtration (LVF).

As we have shown in this study, EVs can be concentrated from high-volume culture supernatants on dialysis membranes (MWCO = 1000 (kDa)) by means of the LVF method. We also compared EVs isolated using LVF with those isolated using the most commonly used alternative methods, ultracentrifugation and differential centrifugation [18], and found that both protein and lipid contents were higher in the LVF samples in comparison to the other isolation methods. FTIR spectroscopy was used to assess the quality and repeatability of EV isolation and molecular content analysis.

It has been reported that ultrafiltration is a good alternative to the ultracentrifugation method [18,20,37]. However, the isolation of EVs through filtration could be challenging, mainly due to the need to rinse EVs from the membrane with additional chemical compounds [38], membrane pore plugging resulting in low isolation yields [39], the possible changes in EV morphology caused by the application of high pressure [40] and time-consuming procedures [20]. LVF can solve these problems because of its important advantages. There is no need to wash EVs from the membrane using chemical compounds, as only water is used to rinse additional proteins from the sample. Membrane plugging, prevalent in other methods, was not observed in our method either. In terms of the yield of isolation, LVF is comparable to ultracentrifugation and significantly better than the differential centrifugation method (shown in Figure 5). As compared to the initial sample, EV concentration was 35 times higher in the case of the ultracentrifugation procedure, 22 times higher in the case of LVF and 12 times higher in the case of differential centrifugation.

The diameter of membrane pores, measured by means of ESEM, varied between 20 and 50 (nm). TEM images of LVF samples confirmed the presence of EVs with a minimal diameter of 20 (nm) (TEM, Figure 4), while NTA analysis showed the presence of EVs with a minimal diameter of 66 (nm) (raw data). It can therefore be assumed that even the smallest EVs can be retained in the filtered media. Moreover, we assessed the impact of applied low pressure on EV shape by means of TEM imaging and did not observe any increase in EV shape diversity in the LVF samples (defined in terms of eccentricity and solidity as shape parameters; shown in Figure 4). However, it is important to consider that TEM is a 2D method and the shape of an object shape is analyzed in sectioned samples. This means that the obtained image is a contour and depends on the orientation of the object.

Another advantage of the LVF method is the isolation time. In our method, pressure is used to facilitate filtration and can speed up the filtration process by up to four-fold (for 100 (mL), reducing time from 8 (h), as for the standard methods, to 2 (h) with the addition of additional pressure) [20]. This facilitation, however, is possible only by means of dedicated centrifuges which are very expensive and not readily available.

In order to confirm that both ectosomes and exosomes were present in the isolated EV samples, and in order to confirm it not only by means of the size distribution methods described above, we applied a Western blot to detect specific exosome and ectosome markers: Hsp70, CD81, CD63 and Arf-6 (Figure 7). We found that LVF samples had the highest intensity of Hsp70 and CD63 bands, while the Arf-6 and CD81 bands had similar low intensity in the samples isolated by all tested methods. Ectosome samples are usually enriched in Arf-6, the protein involved directly in the shedding of plasma membrane-derived EVs, but not in exosome biogenesis [24]. In our study, relatively low expression of Arf-6 in comparison to the exosome marker CD63 was observed, showing that exosome proteins dominated in all samples. In contrast, actin, known as an ectosome marker, was detected with high intensity. We can conclude that actin-based contraction is necessary for exosome secretion and contamination with beta-actin is possible in exosome fractions.

Heat shock proteins (Hsp70) are present in a variety of EVs; nonetheless, these proteins are representative for a canonical exosome [11], and so are tetraspanins (CD81, CD63), which are considered to be exosome biomarkers (Category 1 a: non-tissue-specific transmembrane proteins) [9]. Predictably, HUVEC-derived EVs had higher concentrations of CD63 than CD81 proteins. CD63 is primarily an intracellular tetraspanin, mainly found in late endosomal and lysosomal compartments. In endothelial cells, CD63 was identified as a component of Weibel–Palade bodies [41]. In contrast, CD81 is present in the endothelium of early human atherosclerotic lesions [42].

Therefore, we conclude that the samples isolated by LVF contained the highest concentrations of exosomes, while ectosome concentration was similar in all samples regardless of the isolation method used. It has been postulated that additional centrifugation or filtration steps should be used in order to avoid ectosome contamination in the ultracentrifugation isolation method [43]. In this study, we observed that bands attributed to the exosome marker in the differential centrifugation samples were more than ten times less intensive than in the case of samples isolated by LVF. Based on our findings, we also recommend using additional centrifugation at 18,000 (g) in order to remove ectosomes an additional preliminary centrifugation if the goal is to obtain clear exosomal samples in further isolation steps in ultracentrifugation, as well as before concentration by LVF [8,22].

In our study, we used FTIR as a new approach. Previously, it was shown that FTIR is useful as a screening method for resolving EV protein composition and structure (β-sheet) [28]. In the isolated EV samples, we analyzed three strongest peptide peaks: amide I (1652 (cm^−1^)), amide II (1542 (cm^−1^)) and amide A (approx. 3286 (cm^−1^)). The highest intensity of these peaks was measured for EVs isolated by LVF and ultracentrifugation. Protein bands were barely distinguishable in the spectra of samples obtained by differential centrifugation. Like protein bands, typical lipid bands (3076 (cm^−1^), 2959 (cm^−1^), 2930 (cm^−1^), 2869 (cm^−1^), 1450 (cm^−1^)) were clearly visible for the samples isolated by LVF and ultracentrifugation. Additionally, in the LVF spectra, additional peaks appeared at 1309 (cm^−1^) and 1240 (cm^−1^), which can be attributed to amide III (C–N stretching mode of proteins). The presence of the amide III peak may indicate the presence of α-helix structures in the sample.

Moreover, we applied FTIR analysis not only to assess sample quality, but also to measure protein concentration (the amide-to-lipids ratio). Protein concentration correlated with the amide I-to-lipids ratio for LVF and ultracentrifugation samples (Figure 6 and Figure 7). On the other hand, samples isolated by differential centrifugation and ultracentrifugation had similar protein concentrations. Nevertheless, the amide-to-lipids ratio was significantly different. This is probably a result of the low quality of the samples and the low resolution of the FTIR spectra, where the peaks were barely distinguishable, but calculations of the area under the curve and the ratio between these two values were still possible. The FTIR spectra confirm that the LVF method is reproducible, if other possible error-causing factors (temperature, time of processing, preanalytical errors and the human factor) are controlled.

Plasma membrane-derived vesicles are often used as a model system for the biochemical and biophysical investigation of membrane proteins and membrane organization. The most naïve and unprocessed vesicles from eukaryotic cells are produced by mechanical extrusion of anuclear erythrocytes. Such an approach provides a simple and broadly applicable strategy to isolate and concentrate proteolipidic systems similar to plasma membranes [44]. The other strategy, utilizing a hypertonic calcium chloride buffer to obtain large vesicles (several µm in diameter), was used to quantitatively load protein cargo [45]. In clinics, a bioreactor-based, large-scale production of clinical-grade exosomes was established to generate engineered exosomes with the ability to target oncogenic KRAS (iExosomes) [46]. For such clinical applications, cytochalasin B-induced microvesicles (CIMVs) were found to be effective drug delivery mediators [47]. CIMVs differ from naturally released EVs as they are produced without active cargo sorting machinery from numerous cell types, including HUVECs [48]. The LVF method can be easily utilized to rescale EV purification for clinical DDS.

## 5. Conclusions

The LVF method can be recommended as a workflow for EV isolation from conditioned media in high-volume samples. This method is easy, fast and low-cost, allowing for the isolation of both ectosomes and exosomes from high-volume sources, and could be an efficient alternative to commonly applied methods. These characteristics, especially high reproducibility, may lead to the future applications of this method as an isolation protocol in the development of drug delivery systems based on EVs.

## Figures and Tables

**Figure 1 pharmaceutics-12-00872-f001:**
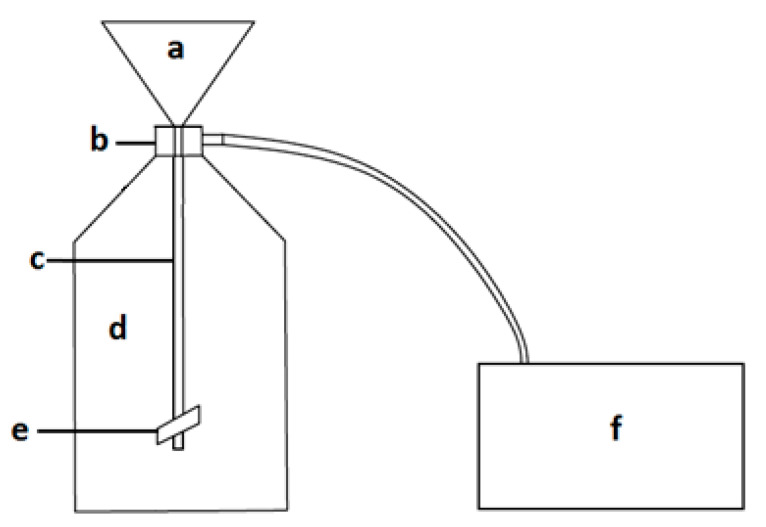
Low-vacuum filtration (LVF) system: (**a**) closed liquid container; (**b**) a coupler connecting the dialysis membrane with the liquid container; (**c**) dialysis membrane; (**d**) a vacuum chamber; (**e**) a clamp closing the membrane; (**f**) a pump.

**Figure 2 pharmaceutics-12-00872-f002:**
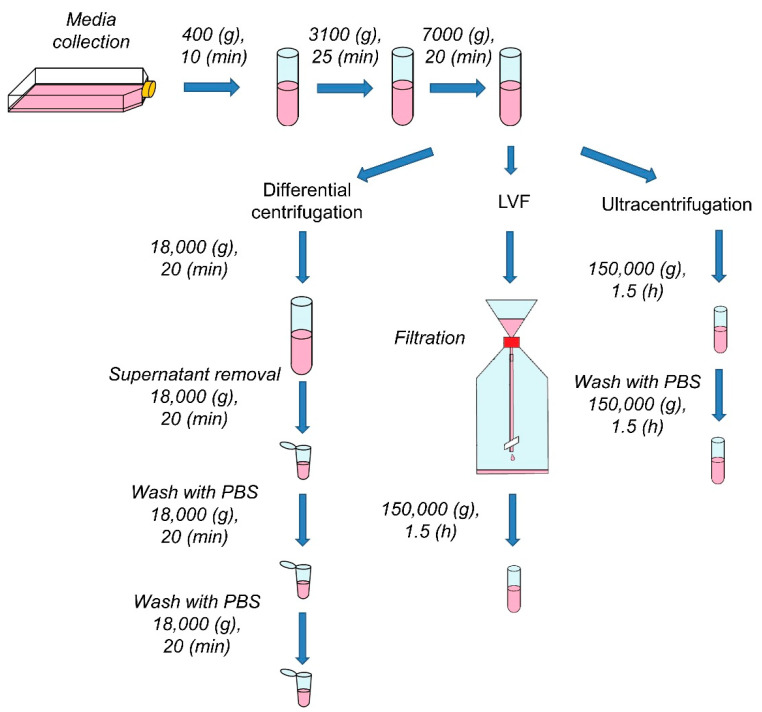
Scheme of the experimental design. After conditioned media collection, the samples underwent preparatory centrifugations to remove cells, cell fragments and apoptotic bodies. After the preliminary steps, EVs were isolated according to the procedure for each method. PBS (phosphate buffered saline).

**Figure 3 pharmaceutics-12-00872-f003:**
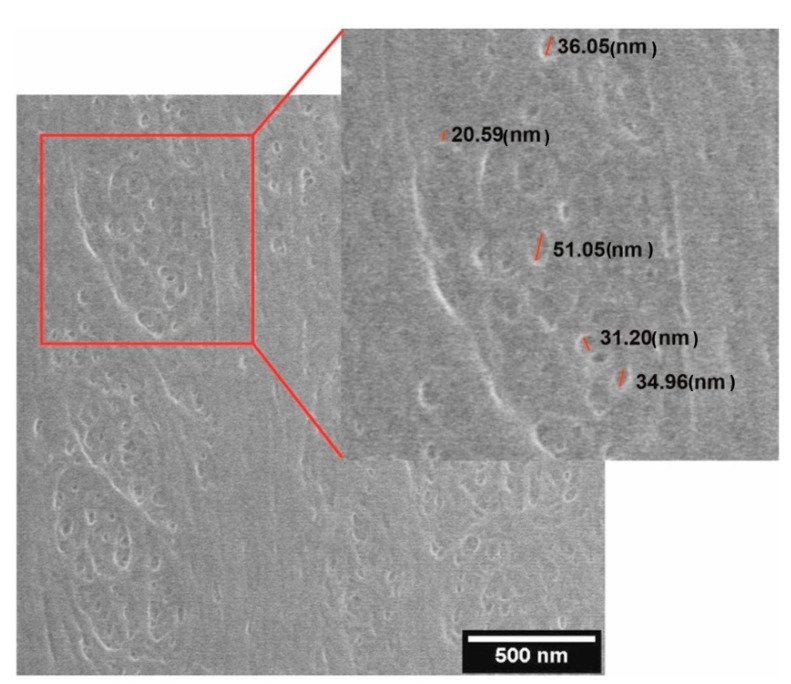
An exemplary Environmental Scanning Electron Microscopy (ESEM) micrograph of the dialysis membrane surface, with the sizes of several pores indicated.

**Figure 4 pharmaceutics-12-00872-f004:**
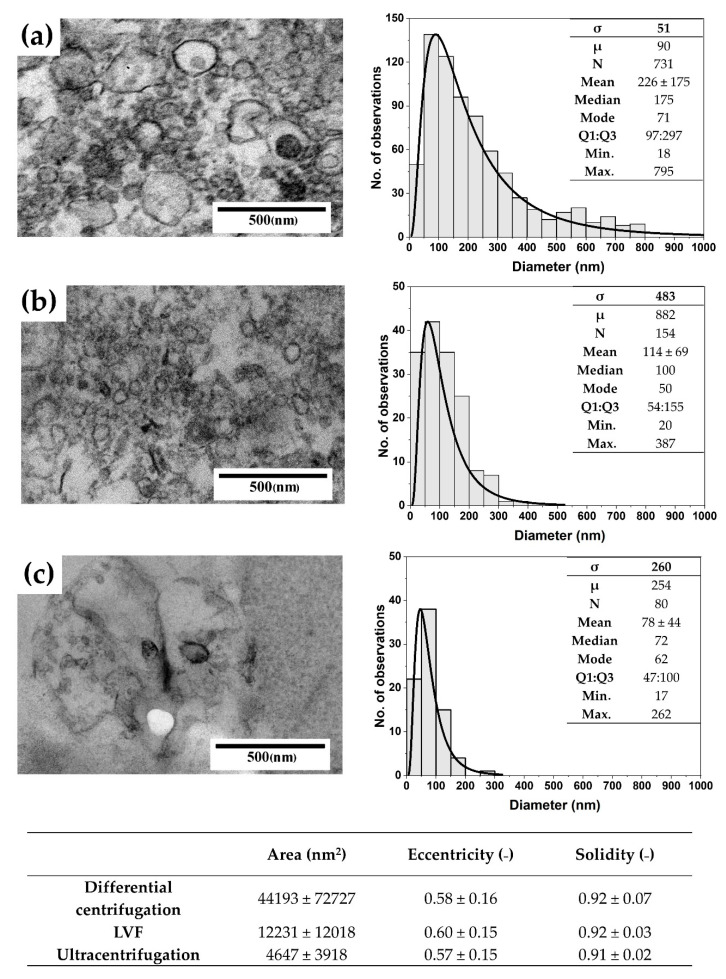
Representative TEM images and size distributions, with the log-normal fit parameters, for the EVs isolated using (**a**) differential centrifugation; (**b**) LVF; (**c**) Ultracentrifugation. The table below the images presents the mean area, eccentricity and solidity of the EVs obtained using these three isolation methods.

**Figure 5 pharmaceutics-12-00872-f005:**
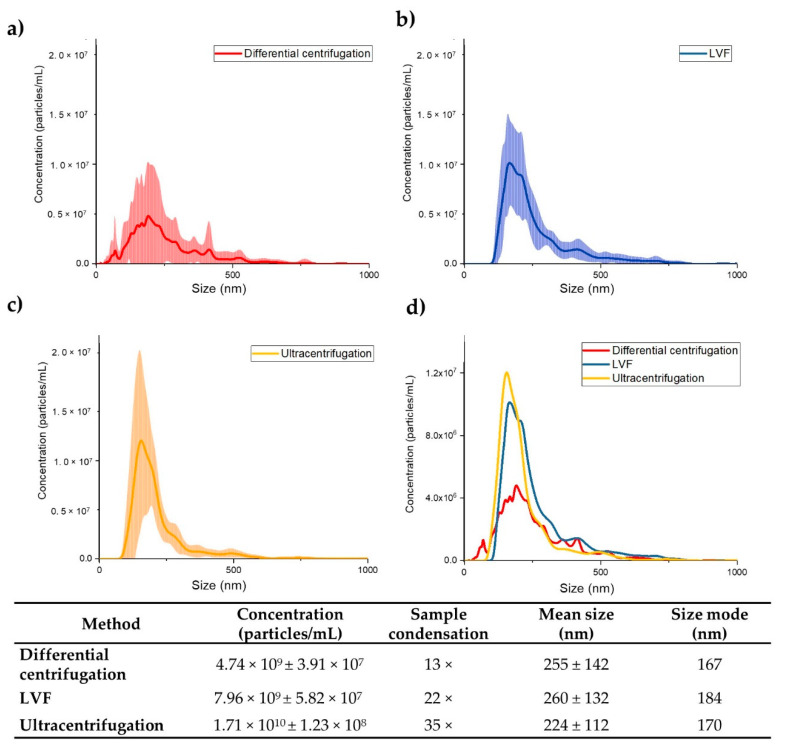
Results of the Nanoparticle Tracking Analysis (NTA). (**a**) Differential centrifugation; (**b**) LVF; (**c**) Ultracentrifugation; (**d**) Comparison of the isolation methods tested. The lines indicate the mean concentrations (line) of the three samples. Each sample was recorded for 30 s and the measurement was repeated 5 times. The SDs of the three independent measurements are presented as the colored areas around the mean concentration line. The table presents the average concentrations of EVs in the samples after isolation; sample condensation, defined as the relative increase in the number of EVs in the sample after isolation compared to their concentration in the starting samples; the average EV diameter; and the size mode.

**Figure 6 pharmaceutics-12-00872-f006:**
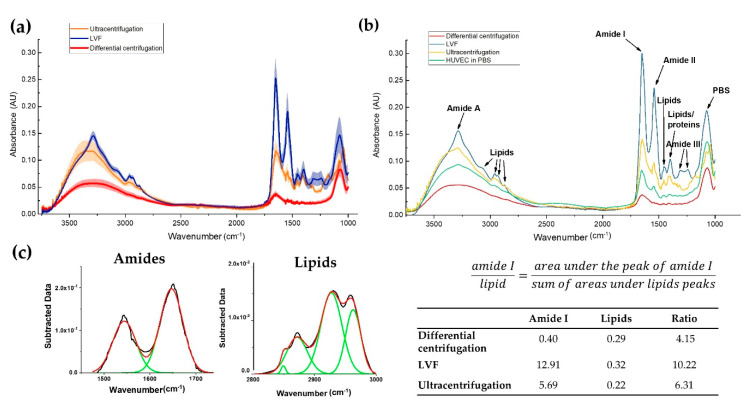
Results of the FTIR analysis. (**a**) Average infrared spectra for EVs isolated with the tested methods (line) with the SD of three independent measurements (upper and lower edges of the colored areas); (**b**) Comparison of the EVs spectra with the spectra of HUVEC cells with the assignment of the main peaks; (**c**) Example of the Gaussian function fitting for amide I/lipids ratio calculation with results (see the table). HUVEC: human umbilical endothelial vein cells.

**Figure 7 pharmaceutics-12-00872-f007:**
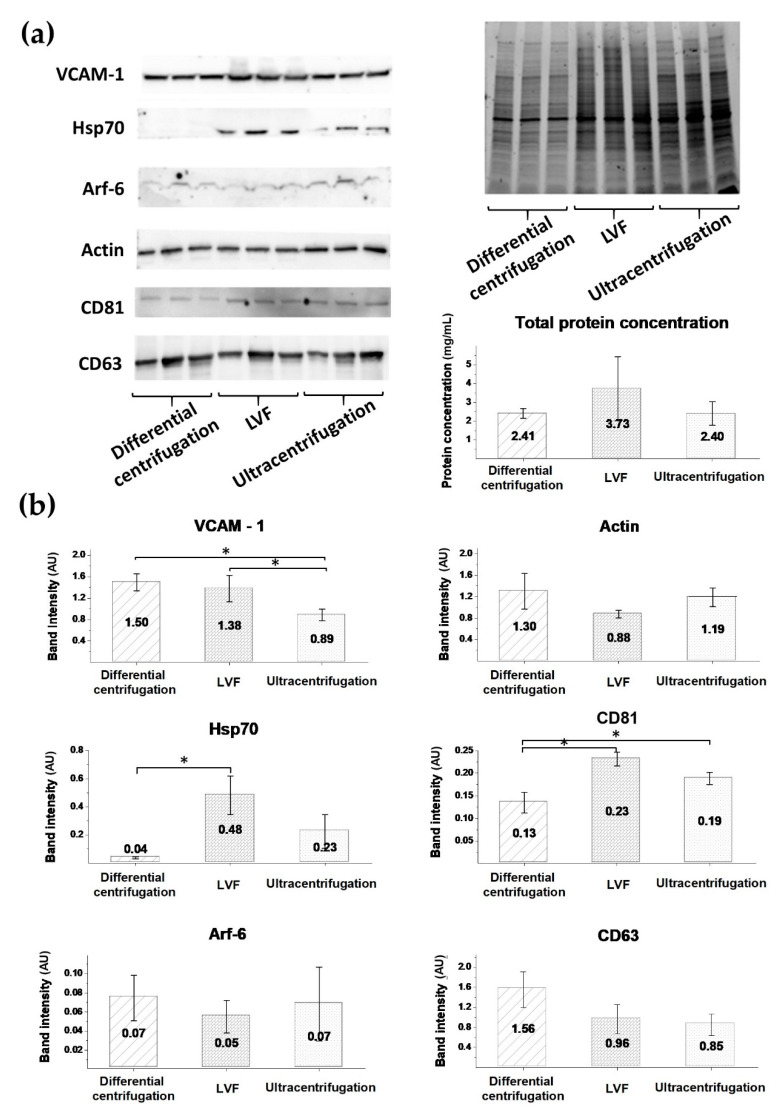
Results of the Western blot analysis. (**a**) Images of the membrane after blotting and the gel after electrophoresis with clearly visible differences in bands’ intensities between the EV samples isolated by different methods; (**b**) Band intensity analysis. Data are presented as mean values (column) and SDs (whiskers). The analysis was performed using the Kruskal–Wallis test. Differences between subgroups were tested with Dunn’s posthoc test, and statistically significant differences are marked with an asterisk (*p* < 0.05).

**Table 1 pharmaceutics-12-00872-t001:** Characterization of EV populations according to diameter, biogenesis, physiological role, cargo and typical markers.

	Exosomes	Ectosomes	Apoptotic Bodies
**Diameter**	30–100 (nm)	100–1000 (nm)	1000–5000 (nm)
**Release mechanism**	Inside the cell in multivesicular bodies	On the cell surface, via blebbing of the cell membrane	Cell fragments generated during cell apoptosis
**Role**	Cell-to-cell communication	Cell-to-cell communication	Phagocytosis facilitation
**Cargo**	DNA, RNA, proteins [9,10]	DNA, RNA, proteins [9,10]	Cell organelles, nuclear fraction [9]
**Markers**	Tetraspanins: CD9, CD63, CD81, Hsp70, Hsp90, Alix, Tsg 101, flotilin [11]	Integrins, selectins, Arf-6	Thrombospondin and C3b

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
