# Peer review of "Low-Vacuum Filtration as an Alternative Extracellular Vesicle Concentration Method: A Comparison with Ultracentrifugation and Differential Centrifugation"

_pharmaceutics, 2020, doi:10.3390/pharmaceutics12090872_

Round 1

Reviewer 1 Report

1.       A significant correction of English is needed throughout the manuscript. 

2.    Obtained data is not sufficient. Additional experiment is an absolute requirement. The authors must characterize EVs isolated by three methods (Low Vacuum Filtration, differential centrifugation, ultracentrifugation) on the presence of CD9, CD63, CD81, actin what are part of the minimal criteria of the international society of extracellular vesicles. Comparison of EVs biological activity or broader analysis of EVs molecular content (using multiplex analysis or WB to detect growth factors, cytokines) is needed.

3.    It would be useful to expand the target audience and discuss the approaches for increasing EVs yield (bioreactor cultures, vesiculation buffer, extrusion through polycarbonate filter, cytochalasin B-induced vesicles): doi:10.1172/jci.insight.99263, doi:10.1021/ac301776j, doi:10.1039/c2ib20022h, doi: 10.1038/s41598-020-67563-9.

Author Response

  1. A significant correction of English is needed throughout the manuscript.

We have corrected grammar errors and we will send the final version for English correction concurrently with this re-submission to have the corrected version for the Editor.

  1. Obtained data is not sufficient. Additional experiment is an absolute requirement. The authors must characterize EVs isolated by three methods (Low Vacuum Filtration, differential centrifugation, ultracentrifugation) on the presence of CD9, CD63, CD81, actin what are part of the minimal criteria of the international society of extracellular vesicles. Comparison of EVs biological activity or broader analysis of EVs molecular content (using multiplex analysis or WB to detect growth factors, cytokines) is needed.

The Authors fully agree with this comment, thus we have performed additional experiments with semi-quantitative western blot analysis to show the EV  homogeneity and exosome content. We have used exosome specific anti-CD63 and CD81 antibodies and anti-actin antibodies to compare tetraspanin content in protein extracts. These experiments are described in the 2.7 section and the 3.5 section (Figure 7.). The results have been commented in the 4 (discussion section).

In this study, the Authors did not tested the biological activity of the isolated EV fractions, it was not the subject of this study and it will be further investigated in the other context of endothelial derived EVs.

  1. It would be useful to expand the target audience and discuss the approaches for increasing EVs yield (bioreactor cultures, vesiculation buffer, extrusion through polycarbonate filter, cytochalasin B-induced vesicles): doi:10.1172/jci.insight.99263, doi:10.1021/ac301776j, doi:10.1039/c2ib20022h, doi: 10.1038/s41598-020-67563-9.

The Authors are grateful for this suggestions, and the Discussion have been expanded and improved and the references [44-47] have been added.

Reviewer 2 Report

The manuscript performed by Anna Drozdz and collaborators addresses the topic of the current methods for EVs isolation from conditioned media and provides a comparison with a new, inexpensive and self-developed method. The text is well organized and structured, English language is correct, and their conclusions and well supported by the data. However, there are some issues that should be addressed:

Abstract: Make sure that all abbreviations are properly explained (ATR...)

Introduction: EVs characterization is still under debate and reaching a common classification is complicated. To solve this tangle, the MISEV 2018 guidelines were published, providing a very extensive and complete characterization of EVs, as well as their classification according to different parameters. I would strongly suggest the author the refer to the MISEV 2018 guidelines for EVs characterization on Table 1. Also, the text and references 8-15 should be deleted as they are not updated in comparison with the MISEV guidelines.

Results: The dialysis membrane pores diameter has an average size of 28.39 +/- 9.63 nm (line 204) and EVs average diameter obtained through LVF (using this dialysis membrane) range 114 +/- 69 nm. Could authors explain this? Specially if they “did not observe any EV shape deformations in the LVF samples” (line 357-358).

If possible, I would appreciate if authors could give further EVs characterization by measuring CD-63 by flow cytometry, electron microscopy, or Western blotting. I suggest performing an immunoelectron microscopy with gold particles that bound to the exosome membrane indicating the presence of the tetraspanin CD-63; this method provides EVs pictures showing their morphology.

Discussion: Please, correct the typing mistake “AFR-6” line 370. As well as the one in line 382 [_B_łąd_!_ _N_i_e_ z_d_e_f_i_n_i_o_w_a_n_o_ _z_a_k_ła_d_k_i_._, B_łąd_!_ _N_i_e_ _z_d_e_f_i_n_i_o_w_a_n_o_ _z_a_k_ła_d_k_i_._].

The Author Contributions paragraph should be revised: remove the template sentences.

Author Response

Abstract: Make sure that all abbreviations are properly explained (ATR...)

This has been corrected and all abbreviations have been explained both in an abstract and in a text body

Introduction: EVs characterization is still under debate and reaching a common classification is complicated. To solve this tangle, the MISEV 2018 guidelines were published, providing a very extensive and complete characterization of EVs, as well as their classification according to different parameters. I would strongly suggest the author the refer to the MISEV 2018 guidelines for EVs characterization on Table 1. Also, the text and references 8-15 should be deleted as they are not updated in comparison with the MISEV guidelines.

A very consolidating paper summarizing the data on the classical and state-of-the-art methods for isolation of EVs, including exosomes and highlighting the advantages and disadvantages of each method has been published in 2018 by Konoshenko MY et al. Isolation of Extracellular Vesicles: General Methodologies and Latest Trends. Biomed Res Int. 2018;2018:8545347. doi:10.1155/2018/8545347, and we have referred this paper in the manuscript [8]

Also the revised MISEV guidelines are referred in the manuscript for EV characterization.

References 8-15 have been deleted.

Results: The dialysis membrane pores diameter has an average size of 28.39 +/- 9.63 nm (line 204) and EVs average diameter obtained through LVF (using this dialysis membrane) range 114 +/- 69 nm. Could authors explain this? Specially if they “did not observe any EV shape deformations in the LVF samples” (line 357-358).

The Authors thank very much for this comment. We read these statements again and we agree that the word “deformations” was not properly used in this context. The idea is that EVs isolated by all 3 methods did not show significant differences in shape parameters. That we slightly deformed (solidity was below 1) and they were asymmetrical (eccentricity was around 0.6) in TEM pictures however the these numbers were not different between methods. One must consider that TEM is a 2D method and an object shape is analyzed in sectioned samples and depends on the object orientation.

In the Discussion section we corrected the appropriate sentence “we did not observe increase in EV shape diversity in the LVF samples” (line 371) and we also discussed the idea that membrane nanopores do not influence shape parameters.

If possible, I would appreciate if authors could give further EVs characterization by measuring CD-63 by flow cytometry, electron microscopy, or Western blotting. I suggest performing an immunoelectron microscopy with gold particles that bound to the exosome membrane indicating the presence of the tetraspanin CD-63; this method provides EVs pictures showing their morphology.

We did western blot analysis for DC63 and Cd81 and results are presented in Figure 7. Because of technical reasons we did not perform suggested immunoelectron microscopy imaging. This technic is now not available for us because of the TEM breakdown, since several weeks and we do not expect repair within next few weeks. We have had closed look into TEM picture of EVs and we find more representative images to showing EV shape and electron density and we disuse in our manuscript (Figure 4).

Discussion: Please, correct the typing mistake “AFR-6” line 370. As well as the one in line 382 [_B_łąd_!_ _N_i_e_ z_d_e_f_i_n_i_o_w_a_n_o_ _z_a_k_ła_d_k_i_._, B_łąd_!_ _N_i_e_ _z_d_e_f_i_n_i_o_w_a_n_o_ _z_a_k_ła_d_k_i_._].

These things have been corrected.

The Author Contributions paragraph should be revised: remove the template sentences

The Authors apologize for this mistake.

Reviewer 3 Report

In their paper entitled “Low vacuum filtration method as an alternative extracellular vesicle concentration method: comparison with ultracentrifugation and differential centrifugation” the Authors set and describe a new method for isolating extracellular vesicles (EVs): Low Vacuum Filtration (LVF). They also compared the new procedure with the purification results obtained with differential centrifugation (DC) and ultracentrifugation (UC).

The paper is of great interest for the Readers of Pharmaceutics: many researchers are indeed evaluating the possibility to use EVs as carriers of therapeutic molecules. However, in order to be used for such an aim, EVs need to be pure as much as possible.

Only a couple of points deserve attention before acceptance:

  1. Completely clear criteria are not yet available , up to now, to distinguish among the different subclasses of EVs; see, for example, 1) Mateescu et al., Obstacles and opportunities in the functional analysis of extracellular vesicle RNA– an ISEV position paper, J Extracell Vesicles. 2017, 6(1): 1286095; and 2) Witwer KW, Théry C. Extracellular vesicles or exosomes? On primacy, precision, and popularity influencing a choice of nomenclature. J Extracell Vesicles 2019, 8(1):1648167. For example, heat shock proteins, such as Hsp70, seem to be present in all kinds of EVs, so, please, consider this point when discussing data and controls (see p. 11, line 370); in general, it should be better to avoid to list in a table the putative characteristics of each subclass;
  2. On page 11, lines 381-382, a notice of error, in Polish language, is present…
  3. Minor: on page 4, line 196, “immunopossitive” instead of immunopositive.

Author Response

In their paper entitled “Low vacuum filtration method as an alternative extracellular vesicle concentration method: comparison with ultracentrifugation and differential centrifugation” the Authors set and describe a new method for isolating extracellular vesicles (EVs): Low Vacuum Filtration (LVF). They also compared the new procedure with the purification results obtained with differential centrifugation (DC) and ultracentrifugation (UC).

The paper is of great interest for the Readers of Pharmaceutics: many researchers are indeed evaluating the possibility to use EVs as carriers of therapeutic molecules. However, in order to be used for such an aim, EVs need to be pure as much as possible.

Only a couple of points deserve attention before acceptance:

  1. Completely clear criteria are not yet available , up to now, to distinguish among the different subclasses of EVs; see, for example, 1) Mateescu et al., Obstacles and opportunities in the functional analysis of extracellular vesicle RNA– an ISEV position paper, J Extracell Vesicles. 2017, 6(1): 1286095; and 2) Witwer KW, Théry C. Extracellular vesicles or exosomes? On primacy, precision, and popularity influencing a choice of nomenclature. J Extracell Vesicles 2019, 8(1):1648167.

The Authors agree with these comment. We finally included the revised MISEV guidelines are referred in the manuscript for EV characterization [9] and added the suggested paper by Mateescu et al. [10]

For example, heat shock proteins, such as Hsp70, seem to be present in all kinds of EVs, so, please, consider this point when discussing data and controls (see p. 11, line 370); in general, it should be better to avoid to list in a table the putative characteristics of each subclass;

The Authors decided  to have this information about specific biomarkers for EV subclasses, and we made some improvements suggested by the Reviewer 2. There is a lot of discussion about appropriate biomarkers use for subpopulation characteristics. In or study we have found that even exosome specific tetraspanins (CD63 and CD81) can differ in the same EV fraction.. (see figure 7). We expected in our study the presence of HSP70 in EV fractions  as representative for canonical exosome (Théry C, Ostrowski M, Segura E. Membrane vesicles as conveyors of immune responses. Nat Rev Immunol. 2009;9(8):581-593. doi:10.1038/nri2567 [11])

  1. On page 11, lines 381-382, a notice of error, in Polish language, is present…

This thing has been corrected

  1. Minor: on page 4, line 196, “immunopossitive” instead of immunopositive.

This thing has been corrected

Round 2

Reviewer 1 Report

Page 11, Line 322: The authors showed that an ectosomal marker Arf-6 were detected with low intensity in all tested samples. At the same time Actin (also ectosomal marker) was detected with high intensity in all investigated samples (Fig.7). The authors should include discussion of this point in Section 4

Author Response

The Authors appreciate very much the reviewer help in the manuscript improvement.

Actin is a common biomarker. Due to its abundance in every cell extract,  actin is used for normalization of expressed proteins. The presence of actin is mostly related with budding vesicles, but also actin was observed in exosome fraction, probably because of contamination with cell proteins. Actin-based contraction is necessary for exosome secretion, thus such contamination is possible [doi.org/10.1083/jcb.201601025]. Exosomes from different cells were characterized and the presence of actin was documented, e.g. exosomes from placental mesenchymal stem cells (pMSC) were enriched with beta-actin [doi.org/10.1371/journal.pone.0068451]. We can assume that b-actin levels in exosomes depends on the secretion activity of paternal cells [doi.org/10.1083/jcb.201601025].

In contrast, ectosome samples are usually enriched in ARF-6, the protein involved directly in the shedding of plasma membrane-derived EVs but not in exosome biogenesis [ref.  24].  In our study showed relatively low concentration of ARF-6 in compare to exosome marker (CD63), showing that exosome proteins dominated in all samples.

We included a short explanation regarding actin abundance in EV fractions.

Reviewer 2 Report

Congratulations, your manuscript is now ready for publication.

Author Response

The Authors appreciate very  much the reviewer help in the manuscript improvement.